# Four-Dimensional Flow Echocardiography: Blood Speckle Tracking in Congenital Heart Disease: How to Apply, How to Interpret, What Is Feasible, and What Is Missing Still

**DOI:** 10.3390/healthcare12020263

**Published:** 2024-01-19

**Authors:** Massimiliano Cantinotti, Pietro Marchese, Eliana Franchi, Giuseppe Santoro, Nadia Assanta, Raffaele Giordano

**Affiliations:** 1Fondazione G. Monasterio CNR-Regione Toscana, 54100 Massa, Italy; cantinotti@ftgm.it (M.C.); pitrino91@gmail.com (P.M.); eliana.franchi@ftgm.it (E.F.); giuseppe.santoro@ftgm.it (G.S.); assanta@ftgm.it (N.A.); 2Istituto di Scienze Della Vita (ISV), Scuola Superiore Sant’Anna, 56127 Pisa, Italy; 3Adult and Pediatric Cardiac Surgery, Department of Advanced Biomedical Sciences, University of Naples “Federico II”, 80131 Naples, Italy

**Keywords:** blood speckle tracking, echocardiography, children, 4D echocardiography, vortex analysis

## Abstract

Blood speckle tracking echocardiography (BSTE) is a new, promising 4D flow ultrafast non-focal plane imaging technique. The aim of the present investigation is to provide a review and update on potentialities and application of BSTE in children with congenital heart disease (CHD) and acquired heart disease. A literature search was performed within the National Library of Medicine using the keywords “echocardiography”, “BST”, and “children”. The search was refined by adding the keywords “ultrafast imaging”, “CHD”, and “4D flow”. Fifteen studies were finally included. Our analysis outlined how BSTE is highly feasible, fast, and easy for visualization of normal/abnormal flow patterns in healthy children and in those with CHD. BSTE allows for visualization and basic 2D measures of normal/abnormal vortices forming the ventricles and in the main vessel. Left ventricular vortex characteristics and aortic flow patterns have been described both in healthy children and in those with CHD. Complex analysis (e.g., energy loss, vorticity, and vector complexity) are also highly feasible with BSTE, but software is currently available only for research. Furthermore, current technology allows for BSTE only in neonates and low-weight children (e.g., <40 kg). In summary, the feasibility and potentialities of BSTE as a complementary diagnostic tool in children have been proved; however, its systemic use is hampered by the lack of (i) accessible tools for complex quantification and for acquisition at all ages/weight, (ii) data on the diagnostic/prognostic significance of BSTE, and (iii) consensus/recommendation papers indicating when and how BSTE should be employed.

## 1. Background

Four-dimensional flow imaging is an evolving cardiovascular imaging technique allowing for revolutionary flow imaging visualization and characterization whose main fields of application are cardiac magnetic resonance imaging (cMRI) and echocardiography [1,2,3,4,5]. Four-dimensional flow echocardiography was introduced in the late 2000s/beginning of the 2010s [4,5,6,7]. The first 4D echocardiographic techniques were contrast-enhanced ultrasound echocardiographic particle image velocimetry (EPIV) (e.g., the speckle tracking of injected microbubbles using contrast echocardiography) [6] and color-Doppler-based vector flow mapping (e.g., a technique combining information on speckles emerging naturally from the blood and color Doppler information on unidirectional flow along the axial axis of ultrasound beam in an angle dependent way) [7]. Potentialities of these preliminary 4D flow echocardiographic techniques were first tested for the study of the direction of septal shunts [7], for the evaluation of vortices within systemic ventricles [4], and for the understanding of complex flow dynamics in stenotic pulmonary valve [5].

More recently (2019), a new echocardiographic technique was introduced [8,9,10,11,12,13,14,15,16,17,18,19]: high-frame-rate blood speckle tracking (BST), using ultrafast ultrasound imaging for blood visualization. BST uses a new non-focal plane wave ultrafast (e.g., 2500–5000 frames per second, reduced on the display to 400–600 frames per second) ultrasound technology [12,13,14,15]. BST offers the advantages of being fast, angle-independent, non-invasive, and very easy to use [12,13,14,15]. BST echocardiography may be helpful in the evaluation of complex flow patterns in congenital heart diseases (CHDs) [12,13,14,15]. BST acquisition requires just a few seconds, like common color Doppler, and imaging re-elaboration is also extremely fast [12,13,14,15]. Compared to conventional color Doppler, BST allows for a more direct and intuitive visualization of complex flow dynamics and for visualization of vortices that are not identified by conventional Doppler techniques [12,13,14,15]. Thus, in complex CHD, the use of BST in conjunction with color Doppler may allow a deeper understanding of the physiology of the cardiac defect, without a significant loss of time [12,13,14,15].

The aim of the current paper is to provide a review and update on potentialities and application of blood speckle tracking echocardiography in children with congenital and acquired heart diseases. Another aim of the present work is to provide tips for the practical use of BST in daily practice.

## 2. Literature Search Criteria

In October 2023, we conducted a review research within three medical engine searches (National Library of Medicine, Science Direct, and Cochrane Library) for Medical Subject Headings (MeSH) and the free text terms “echocardiography”, “blood speckle tracking”, and “children”. The search was further refined by adding the keywords “ultrafast imaging”, “congenital heart disease”, and “4D flow”. In addition, we identified other potentially relevant publications using a manual search of references from all eligible studies and review articles, as well as from the Science Citation Index Expanded on Web of Science.

Titles and abstracts of all articles identified by this search strategy were evaluated, and manuscripts were excluded if (a) the study population involved mixed adult and pediatric populations, (b) used 4D flow techniques different from echocardiography, (c) were focused mainly on an adult non-congenital population, (d) written in a language different from English. (e) All articles were assessed independently by two experts in pediatric echocardiography (M.C. and P.M.) and included in the study after consensus was reached.

## 3. Literature Search Results

### 3.1. Search Results

From 27 publications from registries and 5 from databases, 5 were excluded for duplicate records (n = 2) or marked as ineligible by automation tools (n = 3).

Out of 27 publications identified for potential inclusion into the study, 12 studies were excluded based on the criteria listed above, while 15 were finally selected for analysis and systematic review. Criteria of inclusion and/or exclusion are reported in Figure 1, while in Table 1, major studies on application of echocardiography in children have been listed and detailed.

In the first part of the present review, we will discuss the feasibility of this new echocardiographic technique. Then, we will detail the application of BST to the study of ventricular vortices, and for a deeper understanding of great-vessel flow dynamics. Lastly, potentialities and limitations of BST echocardiography will be discussed.

### 3.2. Feasibility of BST in Children

#### 3.2.1. BST Imaging Acquisition Technique

BST acquisition is like common color Doppler, very fast and totally non-invasive [12,13,14,15]. To acquire a BST movie, it is sufficient going on the color Doppler function, select the BST icon on the screen, and save the image. Re-elaboration of BST saved frames is also very fast and easy. One just needs to press the bottom “show the particles” and the software automatically generates the vortex movie [12,13,14,15]. The whole process (e.g., imaging acquisition and re-elaboration) will take just a few seconds [12,13,14,15]. Good-quality BST images may be acquired in normal conditions, without the need for sedation [12,13,14,15].

#### 3.2.2. Feasibility in Different Conditions

Feasibility of BST in healthy children and in those with different congenital heart diseases has been proven in studies with good sample sizes [11,13,17]. Extremely high (e.g., >99%) feasibility of BST [11] in visualizing flow patterns in the area of interest was firstly proved in 2019 in a study over a mixed population of healthy subjects and fetuses and children with cardiac disease (e.g., 102 subjects, 21 weeks to 11.5 years of age, 4 fetuses, 51 healthy children and 47 children with CHD) [11]. Blood speckle tracking echocardiography furthermore provided accurate for velocity measurements down to 8 cm/s, but compared with pulsed-wave Doppler, BST displayed lower velocities [11]. In another series of 20 infants with CHD [13], BST echocardiography showed its potentialities for a better visualization and deeper understanding of flow dynamics in complex CHD [13] in adjunction with conventional color Doppler. Furthermore [13], it was remarked that BST was highly feasible, reproducible, fast, and easy to use. Other studies [15,18] have proved how feasibility of BST for LV vortex analysis was also very high, varying from 95.6% to 98% [15,18] in healthy children to 93.7% for children with CHD [18]. BST, furthermore, was highly feasible in the evaluation and characterization of aortic flow patterns [8,19], as well as for the quantification of 2D of the left ventricle [15,18] and aortic [19] vortex dimensions. BST offered the advantage of accurate and reproducible quantification of complex and new parameters, such as vorticity [8] and energy loss [8,10], but only with dedicated research software [8,10]. Research software has recently employed (2023) [14] for BST evaluation of left ventricular intraventricular pressure difference (IVPD) in healthy children and in those with different cardiomyopathies with a good feasibility (e.g., feasibility of 88.3% in controls, 80% in children with dilated cardiomyopathy (DCM), and 90.4% in hypertrophic cardiomyopathy) [14].

#### 3.2.3. Summary of Current Evidence

BST analysis is very easy, fast, reproducible, and accurate for blood flow visualization across the heart chambers and main vessels [8,11,13,17,19]. Analysis of left ventricular vortex [15,18] and aortic flow patterns [8,19] is very feasible in both healthy neonates and children [8,15,18,19] and in those with CHD. Two-dimensional quantification of vortexes is feasible with current technology [18,19], while more complex analysis (e.g., energy loss, intraventricular pressure difference, vorticity) [8,14] are feasible, but only with dedicated research software.

### 3.3. BST for the Evaluation of Vortex in Ventricular Chambers

#### 3.3.1. Vortex in the Left Ventricle

Vortices naturally form in all cardiac chambers, but have been studied most extensively in the left ventricle (LV) [12,15,18], where they have supposed to have the function of a reservoir of kinetic energy facilitating systolic ejection of blood flow into the left ventricular outflow tract. The geometry and anatomical locations of vortices are different in healthy adult subjects and in those with cardiac disease [2,3]. Preliminary observations by contrast-enhanced ultrasound echocardiographic particle image velocimetry (EPIV) in a small sample of 9 adults with a Fontan circulation (mean age 31.5 ± 12 years) [4] showed how height and sphericity index of the vortex in the systemic ventricle were significantly smaller and vortex width larger when compared to 15 age-matched controls. The limited data available in children by BST also demonstrate that LV vortex may differ in children with CHD compared to healthy counterparts [19]. A study in 50 preterm infants (weight 500–2020 g) [15] showed how LV vortex area positively correlated with cardiac dimensions including LV diameters (*p* < 0.01), and mitral annulus (*p* < 0.01). In a study [19] of over 60 children with different congenital heart diseases (median age 1.28 years, interquartile range 0.2–6.82 years) and 193 age-matched healthy children, limited differences were noted in vortex distance to apex, distance to interventricular septum, height, width, and sphericity index among CHD and healthy children. Vortex area indexed by body surface area (Vai), however, was significantly higher in children with CHD than healthy subjects (*p* < 0.0001) [18]. Differences in vortex position among different CHDs were furthermore noted in CHD characterized by left ventricle volume or pressure overload associated with vortices localized closer to the interventricular septum [18] Table 2.

Examples of physiological and pathological left ventricular vortices are provided in Figure 2).

Methods for optimal image acquisition have also been provided. A recent (2023) [14] study aiming to validate calculation of the intraventricular pressure difference (IVPD) by BST to assess diastolic suction and early filling compared 31 children with different cardiomyopathies (10 with dilated cardiomyopathy (DCM) and 21 with hypertrophic cardiomyopathy (HCM) with 138 age-matched controls. The authors [14] demonstrated how interventricular pressure difference analyzed during early diastole was reduced in DCM (e.g., −1.21 ± 0.39 mm Hg, *p*< 0.001) and HCM (e.g., −1.57 ± 0.47 mm Hg, *p* < 0.001) compared with controls (e.g., −2.28 ± 0.62 mm Hg), indicating reduced early diastolic suction in these patient groups [14].

#### 3.3.2. Right Ventricle Flow Dynamics

BST echocardiography may be applied not only to the study of the left ventricle but also of the right ventricle (RV), which is often stressed by pressure and/or volume overload in children with congenital heart diseases, particularly those with cono-truncal defects [10,20]. Mawad and colleagues [10] compared right ventricular flow dynamics of 57 children with repaired tetralogy of Fallot (rTOF) with severe pulmonary regurgitation, 11 children with large atrial septal defect (ASD), and 25 healthy controls. Right ventricular diastolic energy loss was similar in repaired tetralogy of Fallot and atrial septal defects, but both were greater than in controls. Locations of high energy loss were also similar in children with repaired tetralogy of Fallot and atrial septal defects, at the level of the right ventricle outflow tract in systole, and around the tricuspid valve in early diastole [10]. An additional apical early diastolic area of energy loss was noted in repaired tetralogy of Fallot, corresponding to colliding tricuspid inflow and pulmonary insufficiency [10] (Table 3).

#### 3.3.3. Summary of the Current Evidence

BST echocardiography may be used for the detection of normal and abnormal vortex formation in the left and right ventricle. Normal, physiological LV vortices are usually smaller than those observed in children with cardiac defects [18]. Data on normal 2D dimensions for LV vortices have been published for neonates and infants (including pre-term neonates) [15,18], while those on RV are still lacking. Data on energy loss [10] and intraventricular pressure [14] may add important additional information to conventional echocardiographic data, but their use is limited to research.

### 3.4. BST for a Deeper Understanding of Aortic and Pulmonary Flow Dynamics

BST may be employed as a complementary tool to color Doppler for a better and more intuitive understanding of flow dynamics across the cardiac chamber and main vessels [11,13]. BST may be helpful to understand the turbulence occurring at the level of the main vessel’s due stenosis, the formation of vortex after the stenosis, and their contribution to the post-stenotic dilatation [5,11,13,16,17]. Similarly, BST may be helpful for a deeper understanding of vortex formation and energy loss in heart chambers due to regurgitant flow [16]. Vortices in the stenotic vessels by vector flow imaging were first described by Honda and colleagues in 2014 in a 15-month-old baby with pulmonary stenosis and post-stenotic dilatation [5]. The authors demonstrated that after percutaneous valvuloplasty, the main pulmonary artery diameter, the vortex dimension, and the energy loss all diminish [5]. More recently, BST has been applied for the evaluation of stenotic and regurgitant semilunar valve disease, with a special focus on aortic disease [8,19]. In Figure 3, we provide an example of vortex propagation in a child with pulmonary stenosis after pulmonary valvuloplasty. BST echocardiography has been also applied to study the characteristic of pulmonary artery blood flow in children with pulmonary hypertension [21]. In 18 children < 10 years with pulmonary hypertension [21], the energy loss vector complexity and diastolic vorticity in the main pulmonary artery were significantly higher in both systole and diastole compared to age-matched controls (e.g., systolic energy loss 4.84 vs. 2.42 mW/m; *p* = 0.01, systolic vorticity 0.21 vs. 0.04, *p* = 0.003; diastolic energy loss 0.69 vs. 0.14 mW/m; *p* = 0.01, diastolic vorticity 0.13 vs. 0.05, *p* = 0.04). Vorticity in the main pulmonary artery (15.2 vs. 4.4 Hz; *p* = 0.001) was also higher in children with pulmonary hypertension compared with controls [21].

#### 3.4.1. Vortex in the Aorta

Physiological flow in the aortic root and ascending aorta is laminar, but not necessarily limited to the axial direction [1,8,19]. Indeed, helical and spiral flow within the aorta have been identified in both healthy and diseased aortas [1,8,19].

The formation of physiological vortices within the aortic sinuses of Valsalva is also well known [22] and was first hypothesized by Leonardo da Vinci [20]. In recent years, adult [3,23] and pediatric [1] cardiac MR studies have been focused on the role of altered high velocity, turbulent, helical flow hemodynamics in post-stenotic dilatation in bicuspid aorta, which is now considered not only a genetic disorder but the sum of a genetic disorder and altered flow dynamics [3,8,11,23,24,25,26,27,28].

Application of BST echocardiography for the evaluation of aortic flow in children is still limited but promising [8]. In 100 healthy children [19], specific patterns of aortic flow by BST were observed. Initially, in early systole, the flow in the aortic root is laminar; however, in mid-systole, the flow splits into two helical branches going in opposite directions (one to the right ascending aortic wall and the other toward the left aortic wall) and vortices form close to the sino-tubular junction [19]. Finally, the flow returns laminar during late systole [19]. This pattern is described in Figure 4.

A study [8] of BST in 14 children with bicuspid aortic valve (BAV) < 10 years of age and 24 age-matched controls revealed how children with bicuspid aortic valve exhibit altered flow dynamics in the aortic root and left ventricle in the absence of significant aortic root dilation (and with mild or no regurgitation and/or stenosis). Children with BAV [8] had on average higher aortic root vorticity (e.g., mean 25.9 Hz, range 23.4–29.2 Hz vs. mean 17.8 Hz, range 9.0–26.2 Hz, *p* < 0.05), vector complexity (e.g., 0.17, 0.14–0.31 vs. 0.05, 0.02–0.13, *p* < 0.01), and rate of energy loss (e.g., mean 7.9 mW/m, range 4.9–12.1 mW/m vs. mean 2.7 mW/m, range 1.2–7.4 mW/m, *p* = 0.01). In children with bicuspid aortic valve, there was higher left ventricular average diastolic vorticity (e.g., 20.9 ± 5.8 Hz vs. 11.4 ± 5.2 Hz, *p* < 0.01), kinetic energy (e.g., 0.11 ± 0.05 J/m vs. 0.04 ± 0.02 J/m, *p* < 0.01), vector complexity (e.g., 0.38 ± 0.1 vs. 0.23 ± 0.1, *p* < 0.01), and rate of energy loss (e.g., 11.1 ± 4.8 mW/m vs. 2.7 ± 1.9 mW/m, *p* < 0.01) [8] Table 4.

Practical examples of differences among physiological and pathological aortic vortices are provided in Figure 5. In Figure 5a, a physiological vortex is shown, while in Figure 5b, an example of a big vortex in the dilated ascending aorta of a child with a bicuspid aortic valve with moderate stenosis has been provided. The propagation of the vortex in the aortic arch in physiological condition and in a child with bicuspid aortic valve with moderate stenosis is shown in Figure 6. In the example, it appears clear that the vortex of the healthy patient (Figure 6a) occupies a smaller area than the pathological vortex and is localized on the ventral surface of the aortic arch, below the first two epi-aortic branches. On the contrary, the vortex of the child with bicuspid aortic valve valvular stenosis (Figure 6b) is located proximal to the epi-aortic branches and occupies the great part of a dilated transverse arch.

#### 3.4.2. Summary of Current Evidence

BST may be employed as a complementary tool to color Doppler for a deeper understanding of normal and abnormal (vorticose) flow patterns across valves and main vessels [8,11,13,16,17,19]. BST may be employed to visualize physiological vortices in the main vessel (e.g., in the aortic root/ascending aorta) and pathological vortex (e.g., those forming after stenosis). Normal [19] and abnormal [8] aortic flow patterns have been described. Data on vorticity and energy loss [8] may add important additional information to conventional echocardiographic data, but their use is still limited to research.

## 4. Current Limitations and Future Direction

Blood speckle tracking echocardiography hold huge potentialities for a new and deeper knowledge of flow dynamics in acquired and congenital heart disease [8,9,10,11,12,13,14,15,16,17,18,19]. Theoretically, application of vortex analysis, vorticity, energy loss [8,10,14] may give us new and powerful instruments to understand the severity of the disease and its progression over time. The systematic and widespread use of BST, however, largely depends on technological evolution. At the moment, licensed software allows only for a limited quantification (e.g., two dimensional measures), while programs for complex analysis (e.g., energy loss, vorticity, inter-ventricular pressure difference, etc.) [8,14,16] have been developed only as prototypes for research. For transthoracic echocardiography, BST acquisition may be obtained only with 12 Hz and 8 Hz probes. This means that BST may be employed only in neonates and low-weight children (e.g., up to 35–40 kg of weight at maximum), thus widely limiting the use of BST, especially in the follow-up of the disease [11,13]. For transesophageal echocardiography (TEE), BST is available only for pediatric probes, but (off-label) its use may be extended also to adults. Lastly, at present, there are no clear recommendations on when or how BST should be employed in a clinical setting. Although there were consistencies among different authors [8,9,10,11,12,13,14,15,16,17,18,19], methods for imaging performance, acquisition (and their optimization), and quantification (for 2D measurements), they have not been standardized yet. For quantification, there is a need for normal values across the spectrum of pediatric age [28], which currently are present only for 2D measures of the left ventricular vortex [12,15,18]. There is a need for wider studies on different CHDs, for better standardization of mechanisms of vortex formation in different conditions, and for their characterization e.g., normal, medium size, big, etc.). Data on energy loss and vorticity also need to be validated in wider studies. The limited data currently available do not allow for a meta-analysis [29,30,31]. The routine use of BST will be feasible only when the technique is available on most of the echocardiographic machines and when basic (2D) and advanced (e.g., energy loss, vorticity, inter-ventricular pressure difference) measures are feasible in a semi-automatic fashion and consistent among different vendors [32]. The evolution of 3D strain echocardiography [33] is crucial also for BST echocardiography for a better understanding of vortex dimensions and their relationship with cardiac structures and great vessels.

## 5. Concluding Remarks

We provided an overview and update of the current application of BST analysis in children with congenital and acquired heart disease, with practical examples and tips for imaging acquisition. Despite being a very new 4D imaging technique, feasibility and potentialities of BST in children with congenital heart diseases have been sufficiently proved. Blood speckle echocardiography is very easy, fast, and reproducible and may be used as a complementary tool to color Doppler for a deeper understanding of normal and abnormal flow and vortex patterns in heart chambers and main vessels. A series of limitations, however, still hamper the systematic use of BST in clinical practice, including (i) the lack of accessible tools for complex quantification (which would provide a clear advantage over conventional techniques), (ii) the possibility to use BST only in neonates and children, (iii) the lack of data indicating normal and abnormal parameters of BST (and their diagnostic and prognostic relevance), and (iv) the lack of standardized methods for performance and quantification of BST analysis. Research data and consensus/recommendation papers indicating when and how BST echocardiography should be employed are required to reinforce and validate the use of BST in routine practice.

## Figures and Tables

**Figure 1 healthcare-12-00263-f001:**
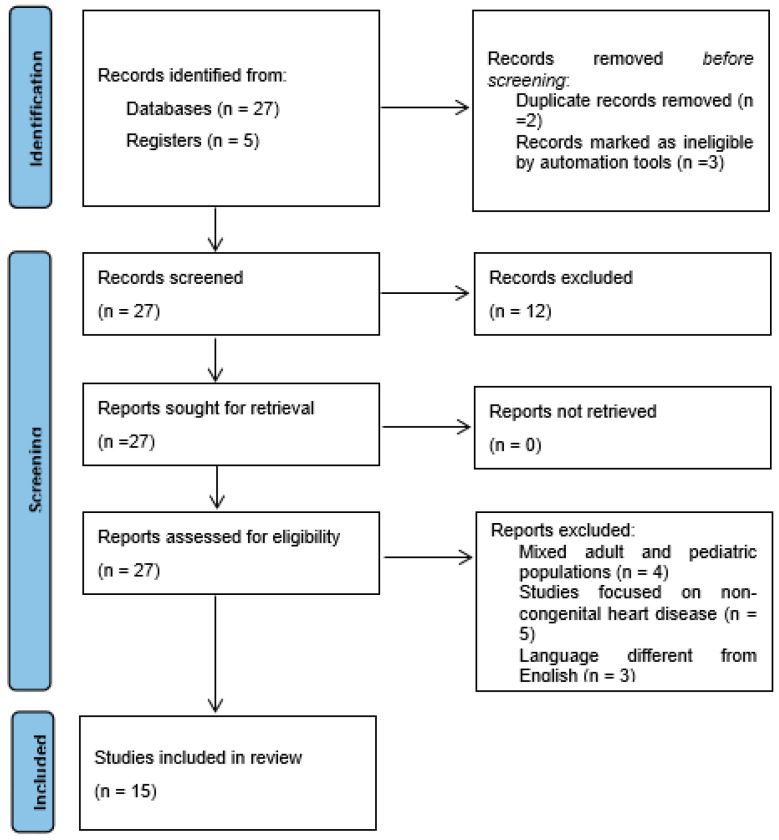
Review flowchart.

**Figure 2 healthcare-12-00263-f002:**
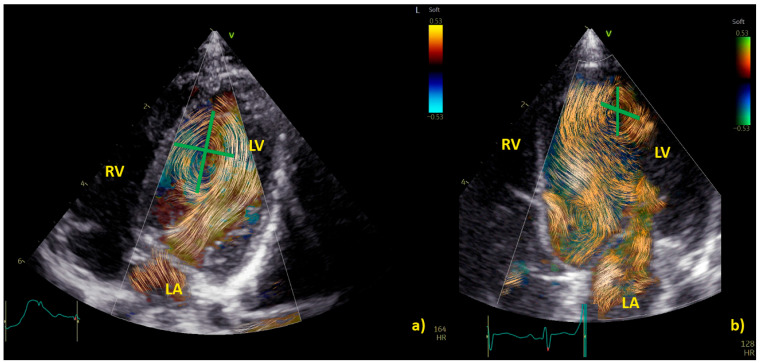
Comparison of left ventricle (LV) diastolic vortex in a healthy heart (**a**) and in a heart with left ventricular volume overload due to a significant patent arterial duct (**b**). BST images were acquired by zoom on an apical 4-chamber view, including both the interventricular septum and the mitral valve. Position of the vortex was calculated in relation to two lines: (1) a line from the ventricular apex to the mitral valve, and (2) a line from the IVS to the LV free wall. Vortex height was determined by measuring the longitudinal dimension of the main vortex relative to LV length, and vortex width as the horizontal dimension of the vortex relative to LV width. The vortex in (**b**) has a different size and position (further from the interventricular septum and with a larger area) compared to the healthy counterpart. LA = left atrium; LV = left ventricle; RA = right atrium; RV = right ventricle.

**Figure 3 healthcare-12-00263-f003:**
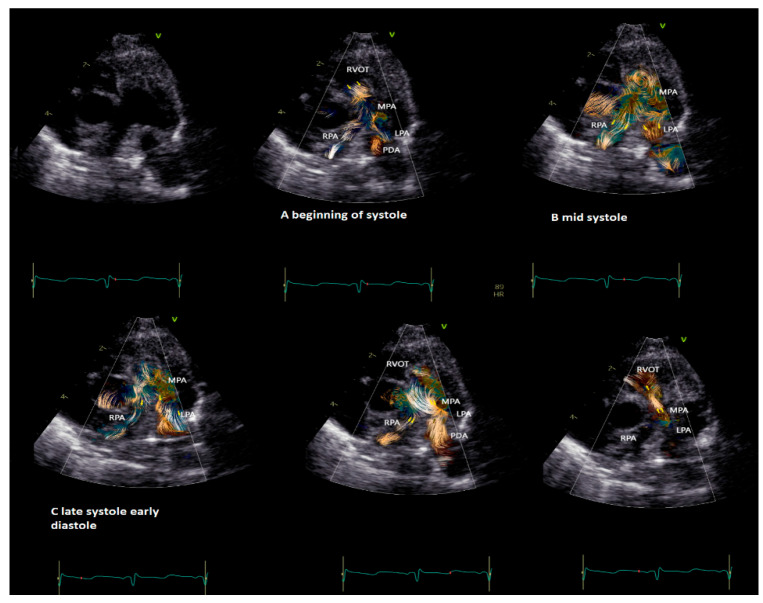
Vortex in the pulmonary valve in a neonate (4.5 kg weight) with pulmonary stenosis after valvuloplasty. The short-axis view shows vortex propagation into the pulmonary arteries during the cardiac cycle. (**A**) The vortex at the level of the pulmonary valve (beginning of the systole), (**B**) propagating into the main pulmonary artery (mid-systole) and into (**C**) pulmonary artery branches (late systole and early diastole). In the last two images, it is possible to observe the flow coming from the patent ductus arteriosus (PDA) that impacts the antegrade pulmonary flow and subsequently the flow coming from the PDA only that directly reaches the RVOT through the insufficiency of the pulmonary valve. The BST movies were acquired at frame rates ranging from 400 to 500 fps in non-sedated infants. Clips of 2 cardiac cycles were stored using medium (66 cm/s), to low (53 cm/s) Nyquist limits.

**Figure 4 healthcare-12-00263-f004:**
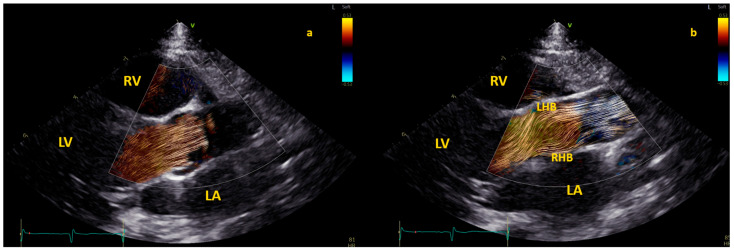
The three different phases of aortic flow in a healthy child. The first phase is shown in the left picture (**a**), where the blood flow is laminar in the aortic root. Both second and third phases are shown in the picture on the right (**b**): double helical flow (with a right–handed branch (RHB) to the left aortic wall, and a left–handed branch (LHN) to the right aortic wall) and laminar in ascending aorta. BST was performed from a focused and zoomed view of the aortic root and the ascending aorta. The color Doppler sector with BST was positioned over the aortic root in the parasternal long-axis view and the ascending aorta in the suprasternal view. Acquisitions included at least 2 cardiac cycles at a range of 380 to 560 frames per second with Nyquist limits ranging from 53 to 58 cm/s.

**Figure 5 healthcare-12-00263-f005:**
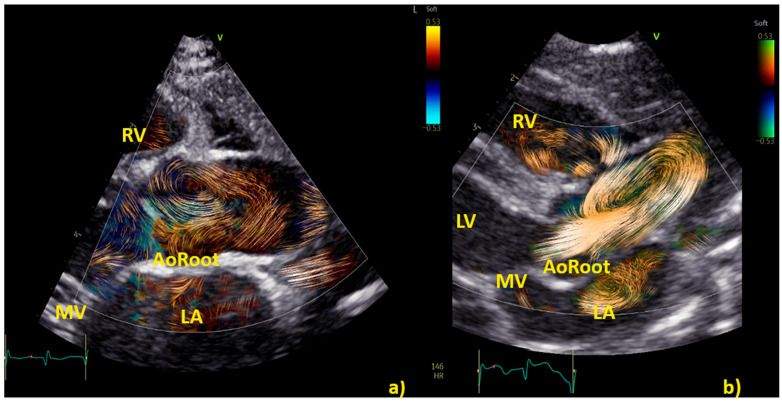
Vortex in the aorta of a healthy child (**a**) and in a patient with a bicuspid aortic valve causing moderate stenosis (**b**) The vortex develops at the end of diastole in both patients, but while in the healthy subject (**a**), physiologically we have a vortex inside the aortic root (AoRoot), in the child with bicuspid aortic valve (**b**), due to the stenosis, the vortex develops more distally, above the sinus-tubular junction, and it is bigger than its healthy counterpart. AoRoot = aortic root, LA = left atrium; LV = left ventricle; MV = mitral valve; RA = right atrium; RV = right ventricle.

**Figure 6 healthcare-12-00263-f006:**
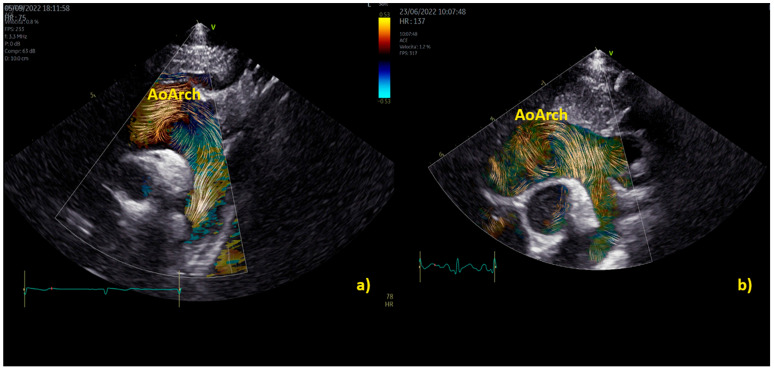
Vortex in the aortic arch in a healthy child (**a**) and in a 3-year-old child with moderate aortic valve stenosis (**b**). It appears clear that the vortex of the healthy patient (**a**) occupies a smaller area than the pathological vortex and is localized on the ventral surface of the aortic arch, below the first two epi-aortic branches; on the contrary, the vortex of the child with valvular stenosis (**b**) is located proximal to the epi-aortic branches and occupies the great part of a dilated transverse arch.

**Table 1 healthcare-12-00263-t001:** Major pediatric studies on BST in children.

Author	Population	Analysis	Software
Nyrnes et al. 2020 [11], Norwegian University and St. Olavs University Hospital Trondheim, Norway	102 subjects (age 21 weeks to 11.5 years)	Feasibility of BST in CHD and reference velocities	Vivid E-9 system (GE Vingmed Ultrasound, Horten, Norway) Research Software
Cantinotti et al. 2021 [13], Massa, Italy	20 neonates and children with CHD	Feasibility of BST in CHD	Vivid E-9 system (GE Vingmed Ultrasound, Horten, Norway)
Borrelli et al. 2021 [16], London (UK) and Padua (Italy)	7 CHD 3 healthy controls	Feasibility of BST in CHD	Vivid E-9 system (GE Vingmed Ultrasound, Horten, Norway)
Marchese et al. 2021 [18], Massa, Italy	193 healthy children (median age 6.33 years, IQR 2.9–10.2 years) 60 CHD (median age 1.28 years, IQR 0.2–6.82 years)	LV vortex characteristics	Vivid E-9 system (GE Vingmed Ultrasound, Horten, Norway)
Mawad W, 2021 [17], Toronto (Canada) and Trondheim (Norway)	57 TF post-repair with severe PR (age 41, 21–74 months) 11 ASD (age 66, 33–99 months) 25 healthy controls (age 42, 26–76 months)	Energy loss in the RV	Vivid E-9 system (GE Vingmed Ultrasound, Horten, Norway) Research Software
Sorensen et al. 2023 [14], Helse Møre and Romsdal (norway) and Toronto (Canada)	138 healthy controls (2 days–17.36 years 10 DCM (0.50–14.92 years), 21 HCM (1.25–17.50 years)	IVPD	Vivid E-9 system (GE Vingmed Ultrasound, Horten, Norway) Research Software
Cantinotti et al. 2023 [19], Massa, Italy	82 healthy children (Age 8.2, 5.6–11.0 years)	Aortic flow patterns	Vivid E-9 system (GE Vingmed Ultrasound, Horten, Norway)
Henry M et al. 2023 [8], Toronto (Canada) and Trondheim (Norway)	38 healthy children (age 1.91, 0.11–5.6 years) 14 BAV (age 4.72, 1.08–7.7 years)	vorticity, vector complexity, energy loss in the Ao and RV	Vivid E-9 system (GE Vingmed Ultrasound, Horten, Norway) Research Software

Ao = aorta, ASD = atrial septal defect, BAV = bicuspid aortic valve, CHD = congenital heart disease, DCM = dilated cardiomyopathy, HCM = hypertrophic cardiomyopathy; IQR = interquartile range, IVDP = intraventricular pressure difference, PR = pulmonary regurgitation, RV = right ventricle, TF = tetralogy of Fallot.

**Table 2 healthcare-12-00263-t002:** Vortex characteristics in healthy subjects and in those with congenital heart diseases. From Marchese P. et al. [18].

	Healthy (n = 118)	Healthy Age-Matched (n = 48)	CHD (n = 43)	P°
Age, years	6.84 (2.94–10.5)	1.53 (0.37–6.84)	0.99 (0.10–6.82)	0.473
Height, cm	121 (89.2–142)	80.0 (62.5–114)	75.0 (58.5–116)	0.613
Weight, kg	25.0 (14.3–37.2)	11.0 (6.55–24.0)	7.95 (4.22–19.0)	0.327
Distance to apex, mm	21.2 (16.0–28.0)	17.9 (13.7–21.0)	16.5 (12.1–23.9)	0.766
Distance to apex/distance from apex to mitral valve, %	39 (32–49)	38 (30–51)	39 (30–48)	0.742
Distance to IVS, mm	11.0 (8.42–13.5)	9.41(6.49–13.4)	8.70 (6.10–13.3)	0.526
Distance to IVS/distance from IVS to LV free wall, %	31 (25–41)	37 (27–45)	34 (25–40)	0.309
Height/BSA, mm/m^2^	10.7 (7.95–15.4)	15.3 (10.3–20.5)	18.6 (11.4–28.3)	0.142
Height/LVEDA, mm/cm^2^	0.75 (0.54–1.31)	0.74 (0.50–1.03)	0.51 (0.40–0.73)	0.429
Width/BSA, mm/m^2^	9.18 (6.86–11.9)	12.4 (9.25–15.6)	16.7 (10.2–22.5)	0.051
Width/LVEDA, mm/cm^2^	0.66 (0.48–0.98)	0.57 (0.44–0.81)	0.43 (0.34–0.57)	0.235
Sphericity Index	1.20 (1.05–1.39	1.24 (1.06–1.50)	1.18 (1.00–1.33)	0.437
Area/BSA, cm^2^/m^2^	0.67 (0.51–0.95)	0.82 (0.63–1.08)	1.01 (0.75–1.64)	0.096
Area/LVEDA, cm^2^/cm^2^	0.03 (0.02–0.05)	0.04 (0.03–0.06)	0.04 (0.03–0.08)	0.349

BSA = body surface area, CHD = congenital heart disease, LVEDA = left ventricle end-diastolic area, IVS = interventricular septum, LV = left ventricle. P° CHD vs. healthy age-matched.

**Table 3 healthcare-12-00263-t003:** Degree and regions of energy loss in healthy children and in those with dilated right ventricles. From Mawad W. et al. [10].

	Diastole	Systole
Controls (n = 25) Age: 42 (26–76) mo Weight:15.8 (12.3–23.6) kg	1.34 (0.55–2.06) (mJ/m) Site: TV	0.17 (0.10–0.48) (mJ/m) Site: RVOT
ASD (n = 11) Age: 66 (33–99) mo Weight: 17.9 (10.5–27.3) kg	2.86 (1.47–3.65) * (mJ/m) Site: TV	0.44 (0.29–0.68) (mJ/m) Site: RVOT
r-TOF (n = 21) Age: 41 (21–74) mo Weight: 12.2 (10.0–19.5) kg	1.93 (1.46–2.74) * (mJ/m) Site: TV, Apical (where TR and PR colloid)	0.29 (0.07–0.51) (mJ/m) Site: RVOT

ASD = atrial septal defect, PR = pulmonary regurgitation, RVOT = right ventricle outflow tract, r-TOF = repaired tetralogy of Fallot, TR = tricuspid regurgitation, TV = tricuspid valve, *p*-value < 0.05: * ASD or rTOF versus CTL.

**Table 4 healthcare-12-00263-t004:** BST aortic parameters in controls and in children with bicuspid aortic valve. From Henry M et al. [8].

	Control (n = 24)	Bicuspid Aortic Valve (n = 14)	*p*
Age (years)	1.91 (0.11–5.6)	4.72 (1.08–7.7)	0.08
Weight (kg)	12.3 (4.0–22.4)	16.85 (7.9–23.9)	0.19
Aortic regurgitation		1 moderate, 5 mild 8 no or trivial	
Peak velocity (m/s)	0.97 (0.8–1.2)	1.9 (1.4–2.5)	<0.001
Peak diastolic kinetic energy (J/m)	0.08 ± 0.05	0.18 ± 0.08	<0.001
Energy loss (mW/m)	6.8 ± 4.7	28.7 ± 15.11	<0.001
Vorticity (Hz)	22.7 ± 7.8	39.8 ± 11.6	<0.001
Vector complexity	0.57 ± 0.18	0.70 ± 0.12	0.02

## Data Availability

Not applicable.

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
