# Peer review of "Four-Dimensional Flow Echocardiography: Blood Speckle Tracking in Congenital Heart Disease: How to Apply, How to Interpret, What Is Feasible, and What Is Missing Still"

_healthcare, 2024, doi:10.3390/healthcare12020263_

Round 1

Reviewer 1 Report

Comments and Suggestions for Authors

1.  A related study/Literature review is required after introduction.

2. "Section 2 . Methods" , need detail discussion with relevant diagrams.

3. " Section 3. Results' , the title should be changed , no specific result of this

       study is there.

4.  Some statistical and mathematical model should be there in the study.

5. "3.3.2 Right ventricle flow dynamics" , need  a diagram to describe the procedure.

6. Detail discussion on BST(Blood speckle tracking) is required in a separate  sub section.

7.  "Section 3.4.1 Vortex in the aorta" , Give a relationship diagram with BST Technology.

8. Need more references from recent publications.

Comments on the Quality of English Language

Through English checking is required.

Author Response

Many thanks for your work. Based on the suggestions raised by the reviewers and editors, we have revised the paper

  1. A related study/Literature review is required after introduction.

Response: we have tried to better clarify literature search criteria

  1. "Section 2 . Methods" , need detail discussion with relevant diagrams.

Response: We added PRISAM 2020 flow chart

  1. " Section 3. Results' , the title should be changed , no specific result of this

       study is there.

Response: we have the change the title as suggested by the reviewer

  1. Some statistical and mathematical model should be there in the study.

Response: we added in the limitation section the difficulty ion doing a metanalysis of current data, due their limited number and heterogeneity.

  1. "3.3.2 Right ventricle flow dynamics" , need  a diagram to describe the procedure.

Response: we added a table as suggested by the reviewer

  1. Detail discussion on BST(Blood speckle tracking) is required in a separate  sub section.

Response: we have detailed more on BST technique in the introduction.

  1. "Section 3.4.1 Vortex in the aorta" , Give a relationship diagram with BST Technology.

Response: we added a table as suggested by the reviewer

  1. Need more references from recent publications.

Response: we added a couple of recent publications

Reviewer 2 Report

Comments and Suggestions for Authors

Consider briefly comparing BST with other imaging techniques in the same domain

The text mentions that BST offers advantages such as being fast, angle-independent, and easy to use, Please elaborate on specific scenarios

The text mentions the use of dedicated research software for certain BST analyses. Please elaborate on the availability, accessibility, and potential future developments in such software. This information could be crucial for researchers and clinicians interested in adopting BST.

It is better to replace the methods section with literature search criteria

Author Response

Many thanks for your work. Based on the suggestions raised by the reviewers and editors, we have revised the paper

Consider briefly comparing BST with other imaging techniques in the same domain

The text mentions that BST offers advantages such as being fast, angle-independent, and easy to use, Please elaborate on specific scenarios

Response 1-2 In the introduction section we have detailed how the imaging acquisition without BST is similar color Doppler and may be used in combination for the evaluation of complex flow patterns. We also provided a comparison between these two tecniques.

The text mentions the use of dedicated research software for certain BST analyses. Please elaborate on the availability, accessibility, and potential future developments in such software. This information could be crucial for researchers and clinicians interested in adopting BST.

Response: we have done it in the limitation section

It is better to replace the methods section with literature search criteria

Response:  we have done as suggested

Reviewer 3 Report

Comments and Suggestions for Authors

This manuscript elegantly summarizes the use of BST in congenital heart disease. The manuscript is well written, I hope the authors take these few remarks into consideration:

Methods: could you provide the exact search terms fo the pubmed search?

Current limitations and future direction: in this paragraph the ideas of the authors on the potential clinical use of BST in the future in lacking. Please elaborate on this in this paragaph

Author Response

This manuscript elegantly summarizes the use of BST in congenital heart disease. The manuscript is well written, I hope the authors take these few remarks into consideration:

Methods: could you provide the exact search terms fo the pubmed search?

Response: we have specified as requested

Current limitations and future direction: in this paragraph the ideas of the authors on the potential clinical use of BST in the future in lacking. Please elaborate on this in this paragaph

Response: we added some “personal ideas” as suggested by the reviewer

Reviewer 4 Report

Comments and Suggestions for Authors

The authors have written a great description of the current literature about blood speckle tracking (BST) in pediatric/congenital heart disease. The field is new, so the literature is limited. This article provides a nice summary of what BST is, how it has been studied, and the differences seen in some types of congenital heart defects compared to healthy controls. This article will be helpful to orient others to the current use and knowledge of this new technology.

A few small suggestions for the figures that will help clarify your presentation:

Figure 2 - You show the measurements of the vortex in 2a. It would be helpful to also show in 2b, if possible.

Figure 3 - because of the width of the image, the individual frames are very small and hard to see without zooming up a lot. If possible, might be better to split it and put 3 frames on top and 3 frames below? You label 3 of the frames A-C and describe in the legend, but it would also be helpful to have description of the other frames. Also, video 5 is mentioned in the figure legend, but not elsewhere and not attached with the manuscript.

Figure 4 - the legend describes the "three different phases of aortic flow", but shows 2 images. If you don't have an image of the last phase, it would be helpful to add a and b to the legend to identify the 2 phases shown and then "not shown" after the final phase.

Comments on the Quality of English Language

A few minor edits are required, but overall the manuscript was well written. Specific areas for edits:

Table 1 - the 1 is missing from 102 in the first row of the population column

Sect 3.2, 1/3 down in first paragraph - "Blood speckle tracking echocardiography furthermore provided accurate velocity measurements down to 8 cm, but compared..." - I think it should be 8 cm/s.

Sect 3.2, last sentence of second paragraph - "research software's" - should be software

Sect 3.3.1 - mid first paragraph - "A study in 50 preterm infants (weight 500 2020 g) (15) showed how LV vortices vortex area positively correlated positively with cardiac dimension including LV diameters (p< 0.01), and mitral annular (p< 0.01)." - take out one of the positively's and changes annular to annulus

Sect 3.3.1 - mid first paragraph, next sentence - "In a study (19) over 60 children..." missing of between study and over

Sect 3.4.1, beginning of 3rd paragraph - "A study (8) by BST over 14 children with bicuspid aortic valve (BAV) <10 years of age" - the "by" should be "of"

Author Response

The authors have written a great description of the current literature about blood speckle tracking (BST) in pediatric/congenital heart disease. The field is new, so the literature is limited. This article provides a nice summary of what BST is, how it has been studied, and the differences seen in some types of congenital heart defects compared to healthy controls. This article will be helpful to orient others to the current use and knowledge of this new technology.

A few small suggestions for the figures that will help clarify your presentation:

Figure 2 - You show the measurements of the vortex in 2a. It would be helpful to also show in 2b, if possible.

Figure 3 - because of the width of the image, the individual frames are very small and hard to see without zooming up a lot. If possible, might be better to split it and put 3 frames on top and 3 frames below? You label 3 of the frames A-C and describe in the legend, but it would also be helpful to have description of the other frames. Also, video 5 is mentioned in the figure legend, but not elsewhere and not attached with the manuscript.

Figure 4 - the legend describes the "three different phases of aortic flow", but shows 2 images. If you don't have an image of the last phase, it would be helpful to add a and b to the legend to identify the 2 phases shown and then "not shown" after the final phase.

Reponses: we have corrected the figures as suggested by the reviewer

Comments on the Quality of English Language

A few minor edits are required, but overall the manuscript was well written. Specific areas for edits:

Table 1 - the 1 is missing from 102 in the first row of the population column

Response: corrected.

Sect 3.2, 1/3 down in first paragraph - "Blood speckle tracking echocardiography furthermore provided accurate velocity measurements down to 8 cm, but compared..." - I think it should be 8 cm/s.

Response: Corrected

Sect 3.2, last sentence of second paragraph - "research software's" - should be software

Response: corrected

Sect 3.3.1 - mid first paragraph - "A study in 50 preterm infants (weight 500 2020 g) (15) showed how LV vortices vortex area positively correlated positively with cardiac dimension including LV diameters (p< 0.01), and mitral annular (p< 0.01)." - take out one of the positively's and changes annular to annulus

Response: corrected

Sect 3.3.1 - mid first paragraph, next sentence - "In a study (19) over 60 children..." missing of between study and over

Response: corrected as suggested

Sect 3.4.1, beginning of 3rd paragraph - "A study (8) by BST over 14 children with bicuspid aortic valve (BAV) <10 years of age" - the "by" should be "of"

Response: Corrected

Reviewer 5 Report

Comments and Suggestions for Authors

Reviewing the manuscript entitled, “4D Flow Echocardiography: Blood Speckle Tracking in Con-genital Heart Disease. How to Apply, How to Interpret, What is Feasible and What is Missing Yet” by Cantinotti M et al., this focuses on a utility of BST in pediatric heart disease. This is a very interesting manuscript and may be important for future non-invasive inspection for CHD. The authors need to respond to my following concerns.

 In the abstract, the authors should distinguish BST from BSTE. The abstract is a complete paragraph and requires explanation of each abbreviation.

 In the introduction section, the authors should describe basic principles of BST echocardiography. Is this inspection non-invasive? Since the target patient is children, it is extremely important whether this is invasive or non-invasive.

 The authors should describe research software dedicated to BST in detail. If this were a non-invasive inspection, I don't think the hurdles for clinical research ethics would be high. BST echocardiography has extremely high clinical utility if blood flow can be measured non-invasively.

 The authors mentioned that it was possible to measure blood flow vortices. The section of 3.4.1. Vortex in the aorta is particularly interesting. Although you mentioned “BST may be employed as a complementary tool to color doppler for a deeper under-standing of normal and abnormal (vorticose) flow patterns across valves, and main vessels (8,11,13,16,17,19).”, why are vortices abnormal? Should I distinguish between vortex and turbulence?

Comments on the Quality of English Language

Minor editing is required. 

Author Response

Reviewing the manuscript entitled, “4D Flow Echocardiography: Blood Speckle Tracking in Con-genital Heart Disease. How to Apply, How to Interpret, What is Feasible and What is Missing Yet” by Cantinotti M et al., this focuses on a utility of BST in pediatric heart disease. This is a very interesting manuscript and may be important for future non-invasive inspection for CHD. The authors need to respond to my following concerns.

 In the abstract, the authors should distinguish BST from BSTE. The abstract is a complete paragraph and requires explanation of each abbreviation.

Response: we have corrected. We have used only BSTE for word limit.

 In the introduction section, the authors should describe basic principles of BST echocardiography. Is this inspection non-invasive? Since the target patient is children, it is extremely important whether this is invasive or non-invasive.

 The authors should describe research software dedicated to BST in detail. If this were a non-invasive inspection, I don't think the hurdles for clinical research ethics would be high. BST echocardiography has extremely high clinical utility if blood flow can be measured non-invasively.

Response: we have clarified that the technique is totally non-invasive. We have added a small paragraph in the feasibility section.

 The authors mentioned that it was possible to measure blood flow vortices. The section of 3.4.1. Vortex in the aorta is particularly interesting. Although you mentioned “BST may be employed as a complementary tool to color doppler for a deeper under-standing of normal and abnormal (vorticose) flow patterns across valves, and main vessels (8,11,13,16,17,19).”, why are vortices abnormal? Should I distinguish between vortex and turbulence?

Response: we have rephrased the sentence by eliminating vorticose that was confusing

Round 2

Reviewer 1 Report

Comments and Suggestions for Authors

1. From figure 2 , briefly explain the activity of  RV , LV and LA.

2. From figure 5 , briefly explain AoRoot.

Comments on the Quality of English Language

Minor revision is required. 

Author Response

Reviewer 2

Comments and Suggestions for Authors

  1. From figure 2 , briefly explain the activity of  RV , LV and LA.

Response: The vortex analysis is performed only ion the left ventricle. Right ventricle and Left atrium are indicated only to orientate the reader within the picture. We have detailed the LV

  1. From figure 5 , briefly explain AoRoot.

Response: we have detailed as requested